# Insights into Disease-Associated Tau Impact on Mitochondria

**DOI:** 10.3390/ijms21176344

**Published:** 2020-09-01

**Authors:** Leonora Szabo, Anne Eckert, Amandine Grimm

**Affiliations:** 1Neurobiology Lab for Brain Aging and Mental Health, Transfaculty Research Platform, Molecular & Cognitive Neuroscience, University of Basel, 4002 Basel, Switzerland; leonora.szabo@unibas.ch (L.S.); anne.eckert@unibas.ch (A.E.); 2Psychiatric University Clinics Basel, 4002 Basel, Switzerland; 3Live Sciences Training Facility, Division of Molecular Psychology, University of Basel, 4055 Basel, Switzerland

**Keywords:** tau protein, mitochondria, tauopathies

## Abstract

Abnormal tau protein aggregation in the brain is a hallmark of tauopathies, such as frontotemporal lobar degeneration and Alzheimer’s disease. Substantial evidence has been linking tau to neurodegeneration, but the underlying mechanisms have yet to be clearly identified. Mitochondria are paramount organelles in neurons, as they provide the main source of energy (adenosine triphosphate) to these highly energetic cells. Mitochondrial dysfunction was identified as an early event of neurodegenerative diseases occurring even before the cognitive deficits. Tau protein was shown to interact with mitochondrial proteins and to impair mitochondrial bioenergetics and dynamics, leading to neurotoxicity. In this review, we discuss in detail the different impacts of disease-associated tau protein on mitochondrial functions, including mitochondrial transport, network dynamics, mitophagy and bioenergetics. We also give new insights about the effects of abnormal tau protein on mitochondrial neurosteroidogenesis, as well as on the endoplasmic reticulum-mitochondria coupling. A better understanding of the pathomechanisms of abnormal tau-induced mitochondrial failure may help to identify new targets for therapeutic interventions.

## 1. Introduction

The brain is a high-energy consuming organ and requires a remarkable 20% of the body’s energy to fulfill its functions. In order to meet this high energy demand, neuronal cells strongly rely on the proper performance of the oxidative phosphorylation system (OXPHOS) of mitochondria [1]. Accordingly, the maintenance of proper mitochondrial function is of utmost importance for functional energy production and consequently the viability of neurons [2]. Thus, long-lasting disturbances may induce various pathologies, ranging from subtle alterations in neuronal function to cell death and neurodegeneration [1]. Therefore, mitochondrial dysfunction seems to be a hallmark of neurodegenerative disorders, such as Alzheimer’s disease (AD), Parkinson’s disease (PD) and Huntington’s disease (HD), and was observed already at early disease stages, before the onset of cognitive impairments [3,4].

Tau protein belongs to the family of microtubule-associated proteins (MAPs) that stabilize microtubule assembly and function (reviewed in [5]). Tau is expressed in most neurons and plays a role in axonal transport, cell polarity, and neurotransmission. Tau is also involved in the pathophysiology of neurodegenerative disorders called tauopathies that are characterized by an aberrant intracellular accumulation of tau within neurons, abnormal tau hyperphosphorylation and assembly into neurofibrillary tangles (NFTs). In primary tauopathies, such as frontotemporal lobar degeneration (FTLD), Pick’s disease, corticobasal degeneration (CBD), progressive supranuclear palsy (PSP), and sporadic multiple system tauopathy, tau plays a primary role and mutant forms of tau proteins were identified [5]. In secondary tauopathies, such as AD, Creutzfeldt–Jakob disease and Chronic Traumatic Encephalopathy (CTE), tau plays also a role in the pathogenesis of the disease but other major factors are involved, as, for instance, amyloid-β (Aβ) accumulation in AD. Substantial evidence has been linking abnormal tau to neurodegeneration, but the mechanisms underlying tau-induced neuronal dysfunction and death are still incompletely understood.

Mounting evidence highlights the dysfunction of mitochondria in tauopathies, including reduced bioenergetics as well as abnormal mitochondrial morphology [6,7]. Besides, abnormal tau has many effects on other cellular functions that may lead to neurodegeneration, which are nicely reviewed elsewhere [8,9,10]. In the present review, we aim to give insights about tau impacts on the different functions of mitochondria in order to draw a “mitocentric” picture of tau toxicity. First, we summarize what is known about tau protein structure and function in health and disease. Then, we remind important aspects of mitochondrial function in order to better apprehend the impact of tau on this paramount organelle. Finally, we discuss how disease-associated tau disturbs mitochondrial functions, including recent developments from the past five years, as well as new insights about abnormal tau effects on mitochondrial neurosteroidogenesis and the endoplasmic reticulum (ER)-mitochondria coupling.

## 2. Tau Protein

### 2.1. Tau Structure and Domains

In the human genome, tau proteins are encoded by a single gene, the microtubule-associated protein tau (*MAPT*) gene, which comprises 16 exons located on chromosome 17q21 [5,11,12,13,14]. Alternative splicing of exons 2, 3 and 10 results in the expression of six different isoforms (2N4R, 1N4R, 0N4R, 2N3R, 1N3R, 0N3R) that are all present in the adult human brain. These splicing variants differ from each other in the presence of zero, one or two N-terminal inserts (0N, 1N or 2N, respectively) and in the number of either three (3R) or four (4R) microtubule-binding repeats in the C-terminal part [10,11,13,15]. While the six isoforms appear to be broadly functionally similar, each is likely to have specific and partially distinctive physiological roles. Of particular interest are the splicing products of exon 10, the 3R and 4R isoforms, normally being expressed in a one-to-one ratio in most regions of mature brains [16,17]. However, deviations from this ratio are associated with certain tauopathies, which can be classified into three groups (3R, 4R or 3R/ 4R) depending on the isoforms found in pathogenic aggregates, and thereby facilitating the onset of the disease [18]. Specifically, compared to 3R tau, 4R tau exhibits a higher affinity for microtubules, and is consequently more efficient in promoting microtubule assembly [10,19]. Interestingly, the two isoforms also seem to impact the motor function of microtubules differentially. Whereas 4R tau decreases the localization of mitochondria to axons to a greater extent than 3R tau, the 3R isoforms are more efficient in increasing the percent of mitochondria moving in the retrograde direction [5,20]. Besides, depending on the isoform tau contains either one or two cysteine residues in the microtubule-binding domain. While in the 3R isoform only C322 within the third repeat is present, 4R tau additionally comprises C291 within the fourth repeat. This variance seems to have an influence on the assembly of paired helical filaments (PHFs) in vitro [21].

The structural basis of tau to bind its interaction partners and to perform its functions lies in the organization of tau’s amino acid sequence. Depending on the biochemical properties, tau can be subdivided into the N-terminal projection domain, the proline-rich region, the microtubule-binding domain and the C-terminal region [22,23,24]. When bound to microtubules, the N-terminal domain of tau projects away from the microtubule surface, where it is believed to interact with components of the neuronal plasma membrane [25,26]. Moreover, this region is involved in determining the spacing between microtubules [11,22,27]. The proline-rich region instead harbors PXXP motifs that provide potential recognition sites for SH3 domain-containing proteins of the Src family kinases, such as Fyn, playing a role in signal transduction [22,26,28]. Furthermore, the ability of tau to interact with microtubules is mediated by the microtubule-binding domain in combination with the adjacent proline-rich flanking domains. Whereas the microtubule-binding repeats bind only weakly to microtubules (but possess specificity for microtubule assembly), the proline-rich region provides an efficient targeting to the microtubule surface [29,30].

Tau is a highly soluble, natively unfolded protein that maintains a highly flexible conformation and overall has little secondary structure [31]. However, when binding to interacting proteins and partners, tau may form local conformations [29]. Furthermore, it has been proposed that soluble tau preferentially changes its global conformation to a paperclip structure, where the C-terminal region folds over the microtubule-binding domain and the N-terminal region folds back over of the C-terminal one, placing them in close proximity [32]. This paperclip fold formation might protect tau from aggregation [33]. Truncated tau has a higher tendency for aggregation [34], which could be probably due to the disruption of the paperclip structure [33]. 

### 2.2. Post-Translational Modifications

Besides alternative splicing, tau is a subject of numerous post-translational modifications that highly regulate the functions of tau in both physiological and pathological conditions. The most commonly described post-translational modification of tau is phosphorylation. Tau is a phosphoprotein containing 85 potential phosphorylation sites on the longest tau isoform consisting of 45 serine, 35 threonine and 5 tyrosine residues [5,35,36]. Tau phosphorylation is developmentally regulated with fetal tau experiencing higher levels of modification than adult tau [37]. The normal phosphorylation state of tau is a consequence of the dynamic regulation between the activities of a large number of protein kinases and phosphatases [22,38]. Many different kinases have been demonstrated to be involved in the site-specific phosphorylation of tau. Tau phosphorylating kinases include, among others, the glycogen synthase kinase (GSK) 3α/ β, cyclin-dependent kinase 5 (Cdk5), mitogen-activated protein kinases (MAPKs), tau-tubulin kinase 1/2 (TTBK1/ 2), cAMP-dependent protein kinase A (PKA), protein kinase C (PKC), 5′ adenosine monophosphate-activated protein kinase (AMPK), calcium/ calmodulin-dependent protein kinase II (CaMKII), and finally tyrosine kinases of the Src family like Src, and Fyn [39]. Of those, GSK 3β and Cdk5 are especially supposed to play a relevant role in the pathogenesis of tauopathies like AD, contributing to increased phosphorylation [23,40]. Conversely, several protein phosphatases including protein phosphatase 1 (PP1), PP2A, PP2B, PP2C and PP5 have been implicated in tau de-phosphorylation [38]. Among them PP2 is considered as the main phosphatase, accounting for approximately 70% of the total tau phosphatase activity in the brain. Moreover, it was demonstrated that PP2 activity is 50% decreased in AD brains, thus leading to increased phosphorylation of tau [41]. While tau phosphorylation traditionally has been the most intensively studied post-translational tau modification, tau is likewise a target of many other post-translational modifications, including acetylation, glycosylation, glycation, deamidation, prolyl-isomerisation, nitration, sumoylation, methylation, ubiquitination, and truncation (reviewed in [5,22]). Together, these modifications differentially regulate the functions of tau and may as well influence one another. Nevertheless, in contrast to the more investigated role of phosphorylation in tau pathology, the implication of the other tau post-translational modifications is yet to be fully characterized.

### 2.3. Physiological Functions of Tau

Tau is considered as a multi-functional protein that plays a number of different roles in neuronal cells. In the adult human brain, tau is predominantly distributed in the axons of neurons [11,42]. There, one of tau’s primary functions is to bind microtubules, where this interaction promotes microtubule assembly and thereby modulates their stability [5,43]. Remarkably, physiologically more than 90% of tau is attached to microtubules [44]. This ability of tau to bind microtubules is mediated by KXGS motifs within the microtubule-binding domain and proline-rich flanking domains [45,46]. While microtubule-binding repeats attach to the inner face, the flanking regions interact with the surface of microtubules [47]. Under physiological conditions, binding of tau to microtubules is a highly dynamic process that is dependent on several factors, including tau isoforms, mutations, post-translational modifications, but also the method used to determine the interaction between tau and microtubules [23]. Concerning the influence of post-translational modifications, this binding ability is most prominently regulated by tau’s phosphorylation state [35,48]. Kinase-mediated phosphorylation of tau detaches the protein from microtubules and subsequently causes their depolymerization, whereas phosphatases de-phosphorylate tau and retain the binding ability to microtubules [49]. Especially the phosphorylation of the KXGS motifs within the microtubule-binding domain has been shown to strongly reduce the binding ability of tau to microtubules [36,50]. Frequent cycles of binding and detachment of tau from microtubules are not only important for the regulation of microtubule stability, but consequently affect the maintenance of effective axonal transport [36]. Axonal transport is a critical process in neurons required for the efficient movement of organelles, lipids, proteins, nucleic acids, and synaptic vesicles. Microtubules provide the platforms for proper intracellular transport by allowing motor proteins to interact with them [51]. While kinesins transport cargoes in the anterograde direction, dyneins are carrying cargoes in the retrograde direction [52]. Upon microtubule binding, tau is involved in the regulation of axonal transport, where tau modulates the motility of kinesin and dynein. When encountering microtubule-bound tau dynein tends to reverse direction, whereas kinesin detaches at patches of bound tau in a concentration- and isoform-dependent manner [53]. Moreover, tau binds to the p150 subunit of dynactin, which stabilizes the interaction of dynein with microtubules, and thus supports dynein-dependent axonal transport [54]. Since axonal transport of cargoes, like mitochondria, to different parts of neurons is essential for a proper synaptic function, pathological changes of tau may lead to the impairment of this transport [55]. In addition to these well-known functions, tau was also found to interact with the actin cytoskeleton. Tau protein may bind to filamentous actin to induce aligned bundles of actin filaments, therefore modifying the organization of the cytoskeleton network [5,56]. To note, physiologically a small amount of tau distributes as well in dendrites. However, the physiological function of dendritic tau has not been well elucidated. Nonetheless, it has been proposed that tau has a dendritic role in the post-synaptic targeting of the Fyn kinase [57,58], where it binds Fyn through PXXP motifs within its proline-rich region and thus promotes the recruitment of Fyn to NMDA receptors [58,59]. The targeting of tau to the post-synapse may play a role in mediating synaptic plasticity, especially long-term depression [60] and hence memory formation [61]. Besides, in axons and dendrites, tau has additionally been detected in the nucleus. Nuclear tau appears to be involved in protecting the integrity of genomic DNA, nuclear RNA and cytoplasmic RNA, thus ensuring their functionality and longevity [62,63,64].

### 2.4. Pathological Aggregation of Tau

Under pathological conditions, alterations in the properties of tau may lead to its aggregation that is characteristic of several neurodegenerative diseases. The conversion of physiologic soluble tau species into pathologic fibrillary tau aggregates is considered to be a multi-step process [11]. As one of the mechanisms to drive tau aggregation aberrant phosphorylation has been assumed, since hyperphosphorylation and aggregation of tau are both increased in AD [65]. Hyperphosphorylation of tau is most likely to result from an imbalance in the activities of specific tau kinases and phosphatases, causing an increased rate of tau phosphorylation and/ or decreased rate of de-phosphorylation [66]. Consequently, tau hyperphosphorylation reduces its binding affinity to microtubules, thereby induces a loss of tau’s normal microtubule-stabilizing function [67,68], and thus causes microtubule depolymerization [49]. Specifically, the phosphorylation of KXGS motifs within the microtubule-binding domain (in particular S262) and S214 within the flanking region of tau have been described to strongly decrease the affinity of tau for microtubules [69,70]. Furthermore, in vitro studies demonstrated that phosphorylation of T231 within the flanking region also contributes to the reduced binding of tau to microtubules [71]. The detachment of tau from microtubules subsequently leads to an abnormal increase of free unbound tau in the cytosol [72]. This higher cytosolic concentration may render tau substantially more likely to undergo misfolding. Thereafter, as an early pathological event, non-fibrillar tau deposits, referred to as pre-tangles, are formed. Following steps comprise conformational changes leading to the generation of PHFs. This transition from pre-tangles to PHFs includes the formation of characteristic β-sheet-like structures [11,73]. Precisely, the hexapeptide motifs PHF6 and PHF6* located in the second and third microtubule-binding repeats exhibit a high β-sheet propensity, and are supposed to promote abnormal tau aggregation in vitro and in cell and animal models [22,23,26]. Finally, PHFs further self-assemble to form more organized aggregates, and eventually develop insoluble NFTs inside neurons. The following sequestration of NFTs together with compromised cytoskeleton dynamics impairs normal axonal transport, and hence contributes to synaptic dysfunction and neurodegeneration [74,75]. In addition, alterations of tau itself, such as mutations in the *MAPT* gene, can also contribute to tau aggregation. For instance, in the tau mutations P301L, P301S and ΔK280 that are found in frontotemporal dementia with parkinsonism-17 (FTDP-17) the hexapeptide motif PHF6* is present. As a result of this enhanced β-sheet propensity, tau with these mutations tends to have a decreased affinity for microtubules and an increased ability to assemble into filaments, thus promoting tau aggregation [22,30,76]. To point out, even though phosphorylation of S262 and S214 strongly prevents the attachment of tau to microtubules, phosphorylation of these sites tends to inhibit PHF formation [21].

Although tau phosphorylation is frequently considered as one of the most important modifications leading to aggregation, emerging evidence has related N-terminal truncated tau to tau pathology. In fact, several specific truncations of tau have been identified in AD brains. Moreover, studies demonstrated that proteolytic cleavage of tau at N368, D421 and E391 increases its susceptibility to form NFTs [33,77]. Despite intense investigation, the precise pathogenesis of tau-mediated neurodegeneration in tauopathies still remains unclear. Although the accumulation of insoluble aggregated tau deposits in form of NFTs is considered as a pathological hallmark of tauopathies [78] with their regional distribution correlating with the severity of the cognitive decline in AD brains [79], the neurotoxicity of NFTs per se is controversial. Instead, recent evidence indicates that small soluble oligomeric forms of tau, generated during tangle formation, are the most toxic tau species causing neuron damage and synaptic dysfunction [80]. However, the toxic gain of function by NFTs might contribute to the disease progression as well [16]. Of note, increasing evidence has been linking tau pathology and neuroinflammation. Indeed, abnormal tau was associated with reactive microglia, as well as increased levels of pro-inflammatory cytokines (e.g., interleukin-1β) and complement proteins (reviewed in [81]). The chronic neuroinflammation may lead to synapsic loss and cognitive decline. Interestingly, microglial cells themselves seem to be involved in the spreading of tau pathology [82].

Current therapeutic strategies targeting tau consist of anti-aggregation agents (regulation of tau phosphorylation, inhibition of tau aggregation), tau passive immunotherapy, tau therapeutic vaccines, targeting of tau gene expression (antisense therapies) and therapeutic reduction of tau (reviewed in [83]). Strikingly, the effects of tau-targeting drugs on mitochondrial function remain underinvestigated. Conversely, evidence showed that improving mitochondrial quality control via activation of mitophagy (removal of damaged mitochondria, see Section 3.2.) decreases tau pathology in different experimental models (reviewed in [84]). Namely, nicotinamide riboside supplementation (activator of mitophagy) was shown to decrease abnormal tau phosphorylation, neuroinflammation and cognitive impairments in AD transgenic animals [85,86]. Of note, mitophagy itself seems to play also a role in the regulation of the inflammatory response [87]. Further studies need to be performed to unravel the role of mitophagy in the reduction of tau pathology via modulation of neuroinflammation.

To our knowledge, only one of the tau-targeting agents undergoing clinical trials was shown to also have an effect on mitochondria [88]. Indeed, methylene blue (MB), also known as methylthionine chloride (MTC), is already an approved drug against malaria, and acts as a direct inhibitor of tau protein aggregation [83]. MTC was shown to counteract oxidative stress-induced mitochondrial damage, and to inhibit the monoamine oxidase A that is a source of reactive oxygen species (ROS) [88]. Despite the improvements observed in AD-related symptoms during phase II clinical trials, too many undesirable side effects were reported (e.g., dizziness, diarrhoea, painful urination) for the drug to be used (Clinical Trial Identifier, NCT00515333 and NCT00684944) [83].

## 3. Mitochondria

Mitochondria are complex cytosolic organelles in eukaryotic cells that have been known for over a century. Unlike other organelles, they are maternally inherited (although biparental inheritance of mitochondrial DNA was recently reported but still under debate [89,90]) and compartmentalized. Concerning their structural characteristics, mitochondria consist of a matrix and two membranes with an interjacent intermembrane space (IMS). The outer mitochondrial membrane (OMM) smoothly envelops and separates the organelle from the cytosol, whereas the inner mitochondrial membrane (IMM) is tightly folded, forming multiple invaginations termed as cristae. These organelles play a pivotal role in cell survival and death by regulating both energy metabolism and apoptotic pathways. Additionally, they contribute to an array of cellular functions, including intracellular calcium homeostasis, reduction-oxidation (redox) signaling, innate immunity, steroid biosynthesis and synaptic plasticity, to name a few [91,92]. In the following paragraphs, we will introduce key aspects of mitochondrial physiology that are relevant to understand the deleterious impact of abnormal tau on this paramount organelle (described in Section 4 and Section 5).

### 3.1. Mitochondrial Bioenergetics

Despite this diversity of functions, mitochondria remain best known as the main source of cellular energy production in the form of adenosine triphosphate (ATP) via OXPHOS. OXPHOS refers to the metabolic process in which electrons are transferred stepwise to oxygen through a series of redox reactions between protein complexes to ultimately drive the synthesis of ATP. These protein complexes are embedded in the IMM and comprise four biochemically linked multi-subunit complexes (Complexes I, II, III and IV), known as the electron transport chain (ETC), and the ATP synthase (Complex V) [93]. Briefly, enzymes of the tricarboxylic acid cycle within the mitochondrial matrix oxidize acetyl-CoA, which is derived from carbohydrates, fats and proteins, to generate the reducing equivalents nicotinamide adenine dinucleotide (NADH) and flavin adenine dinucleotide (FADH_2_) [94]. Subsequently, these molecules pass their electrons to the ETC. The following oxidation of these reduced substrates causes conformational changes in the respiratory chain Complexes I, III and IV, allowing them to pump protons (H^+^) out of the mitochondrial matrix into the IMS [95]. This in turn, produces a proton-gradient and thereby establishes an electrochemical potential, termed as mitochondrial membrane potential (ΔΨm) [2]. As a result, Complex V utilizes this proton motif force to catalyze the synthesis of ATP by phosphorylating adenosine diphosphate (ADP) [96] (Figure 1).

Although the physiological functions of mitochondria, such as the production of energy, are critical for cell survival, they also induce the formation of ROS that can pose serious damage to cells when generated in excess [1]. In order to avoid exceeding levels of ROS, cells possess efficient antioxidant mechanisms that scavenge ROS to non-toxic forms [97]. Consequently, under physiological conditions, the production and detoxification of ROS are balanced. However, an increased ROS production and/ or a reduced antioxidant system can induce oxidative stress, which in turn damages proteins and DNA, and initiates lipid peroxidation. Since long polyunsaturated fatty acids chains of mitochondrial membranes exhibit a high susceptibility to oxidation, mitochondria represent the first targets of ROS toxicity, leading to depolarized membranes and damaged proteins, and consecutively to mitochondrial impairments. As a result, oxidative stress-induced mitochondrial dysfunction may initiate cell death, and has been implicated in the pathogenesis of many neurodegenerative diseases, including AD [1,98,99]. Namely, mitochondria-derived oxidative stress was proposed to be a causative factor for Aβ and tau pathology [1,98,100]. Indeed, Aβ load was increased in cells and mice that produced more ROS due to a mitochondrial Complex I inhibition/ deficiency [101]. Similarly, increased levels of tau and tau phosphorylation (at S396, S404, T205, T231) were observed in mice lacking the detoxifying enzyme superoxide dismutase 2 (SOD2) [102]. This suggests that mitochondrial dysfunction, more precisely mitochondria-derived ROS, might be involved in the pathogenesis of tauopathies.

### 3.2. Mitochondrial Dynamics

As commonly known, mitochondria are highly dynamic organelles that form a remarkably complex interconnected network, which varies greatly in different cell types. In response to external and internal stimuli, mitochondria are therefore able to adapt rapidly the degree to which they are networked [103]. Concretely, mitochondrial dynamics results from the interplay of processes, including mitochondrial biogenesis, mitochondrial fusion and fission, mitophagy, and mitochondrial trafficking (Figure 2). These processes maintain the mitochondrial homeostasis and regulate mitochondrial morphology and their distribution, ultimately being a vital component of the cellular stress response [104].

The morphology of the cellular mitochondrial network is sustained by continuous rounds of fusion and fission. Consequently, the balance between these two opposite processes modulates mitochondrial number, shape and size [105]. While increased fusion generates elongated, interconnected mitochondria, enhanced fission promotes mitochondrial fragmentation. This plastic adaptation is particularly essential in neurons that are highly polarized cells. Since axons and dendrites have differential energy demands, the regulation of fusion and fission produces a generally more elongated network in the somatodendritic compartment and a more fragmented one in axons [106]. Notwithstanding, the mitochondrial integrity is fundamental for the mitochondrial metabolic activity and mitochondrial health. For instance, in fused organelles, the efficiency of ATP production is increased and the exchange of matrix content—including mitochondrial DNA—is favored. Fragmented organelles, in contrast, produce more ROS and are readily cleared by mitophagy. Nevertheless, mitochondrial fragmentation is indispensable during cell division for the equal distribution of mitochondria to daughter cells [107].

In mammalian cells, mitochondrial fusion involves the actions of three large dynamin-related guanosine triphosphatases (GTPases). While mitofusin 1 (MFN1) and 2 (MFN2) are the key mediators of the OMM fusion, optic atrophy 1 (OPA1) mediates the fusion of the IMM. Of note, OPA1 also regulates the remodeling of mitochondrial cristae, which is implicated in processes such as apoptosis [108]. Conversely, mitochondrial fission in mammalian cells is primarily orchestrated by dynamin-related protein 1 (DRP1), which is also a GTPase. In order to initiate mitochondrial fission, DRP1 needs to be recruited from the cytosol to mitochondria. This translocation depends on its phosphorylation state at S637. Whereas PKA-mediated phosphorylation of S637 retains DRP1 in the cytoplasm, Ca^2+^-dependent phosphatase calcineurin de-phosphorylation targets DRP1 to the OMM. This has been described to occur preferentially at regions associated with the ER. Subsequently, DRP1 oligomerizes into ring-like structures around mitochondria, providing the required mechanical fission force. As a consequence, guanosine triphosphate (GTP) hydrolysis induces membrane constriction, and thus facilitates scission. A series of additional proteins have been presented to recruit and assemble DRP1, and therefore assisting in the complete separation of mitochondria, including mitochondrial fission factor (MFF), mitochondrial dynamics protein 49 and 51 (MiD49/ 51), and mitochondrial fission 1 protein (FIS1). Although FIS1 was formerly identified as an essential regulator of mitochondrial fission, recent studies demonstrated that it appears to be dispensable for physiological fission. However, FIS1 may be involved in pathological fission processes. Since mitochondrial fusion and fission are crucial for the maintenance of cell survival, a pathological imbalance is associated with many disorders, such as neuropathies, brain injury, extreme stress conditions, aging, and neurodegenerative diseases [105,107,109]. Unsurprisingly, an altered balance of these processes has been reported in AD, both on transcript and protein levels [110,111].

Beyond mitochondrial fusion and fission, another fundamental aspect involved in mitochondrial dynamics includes the regulation of mitochondrial trafficking. The proper subcellular distribution of mitochondria in neuronal cells is indispensable for ATP provisioning, axonal growth promotion, calcium buffering, and to ensure mitochondrial repair and degradation. In mammals, mitochondrial transport is mediated via the activities of the motor proteins dynein and kinesin, which are associated with the adaptor and receptor proteins of the OMM. Precisely, the transmembrane receptor protein mitochondrial Rho GTPase (Miro) interacts with the adaptor protein Milton that in turn tethers to motor proteins. Notably, when production of ATP and calcium buffering are required at specific sites, mitochondrial movement can be halted, and thus mitochondria are retained [112,113]. Accordingly, a loss of this Miro-Milton-dependent transport may cause depletion of mitochondria in dendrites and axons, giving arise to neurotransmission defects [114].

Equally important to fusion and fission, mitochondrial biogenesis and mitophagy regulate the dynamics of mitochondria. The balanced action of these two opposing cellular pathways determines the mitochondrial mass and accurate turnover, and is thus crucial for maintaining a healthy mitochondrial pool. Hence, the tight coordination of mitochondrial biogenesis and degradation is essential for the cellular adaptation in response to the cellular metabolic state, stress, and other intracellular or environmental signals [115,116]. During mitochondrial biogenesis, mitochondria increase their individual mass and copy number in order to either elevate mitochondrial function in general, or to compensate for decreased mitochondrial mass, resulted from higher rates of degradation. Damaged or energy-deficient mitochondria can be selectively degraded via mitochondrial autophagy, a process termed mitophagy. One of the best-characterized mitophagy pathways involves the operations of the proteins PTEN-induced kinase 1 (PINK1) and Parkin. The process of mitophagy is initiated, when dysfunctional fractions of the mitochondrial network cause a loss in ΔΨm. Although PINK1 is normally imported and degraded within the organelle, this depolarization results in the accumulation of PINK1 in the OMM. Accumulated PINK1 has been shown to phosphorylate ubiquitin in the OMM, consequently leading to the recruitment of the cytosolic ubiquitin ligase Parkin to the surface of mitochondria. Following translocation, activated Parkin ubiquitinylates proteins of the OMM. Subsequently, impaired mitochondria are recognized and engulfed into the autophagosome, ultimately targeted for lysosomal degradation [113,117]. Given the pivotal role of mitophagy in maintaining mitochondrial quality control and homeostasis, unsurprisingly, suppression or abnormalities of this process may result in the accumulation of damaged mitochondria. Indeed, mitochondrial dysregulation with regard to mitophagy has been implicated in several neurodegenerative diseases, including PD, AD, and HD. Recent findings in AD patients with sporadic late-onset AD emphasized that mitophagy is compromised, leading to the accumulation of dysfunctional mitochondria, and thus contributing to synaptic dysfunction and cognitive deficits [118].

## 4. Mitochondria: Target of Tau

Abnormal tau impairs mitochondrial function, leading to neuronal degeneration, but the exact mechanisms are still not completely understood. In this section, we will discuss different impacts of abnormal tau on mitochondria in order to draw a “mitocentric” picture of tau toxicity. Noteworthy, nearly all the data discussed here derived from in vitro and animal studies. Therefore, studies performed on patients with tauopathies are highly needed, in order to confirm and fully apprehend mitochondrial dysfunctions induced by abnormal tau protein.

Important mitochondrial impairments observed in the presence of abnormal tau are summarized in Table 1 and Figure 3.

### 4.1. Mitochondrial Transport

Being a member of the family of MAPs, tau is involved in the transport of cargoes along the axons, including mitochondria. In K3 mice expressing the K369I tau mutation, anterograde axonal transport of mitochondria was impaired, reducing the number of mitochondria at the synapse [119]. Synaptic mitochondria play important roles in calcium buffering and fulfill the high energy required in this cellular compartment. Therefore, a decrease in mitochondrial transport to the synapse may lead to synaptic degeneration and neuronal death [106]. One proposed mechanism is that abnormal tau interacts with c-Jun N-terminal kinase-interacting protein 1 (JIP1), which is associated with the kinesin motor protein complex [134]. By sequestrating JIP1 in the cell body, abnormal tau impairs its transport to the axon, which disturbs the formation of the kinesin motor complex and impacts the kinesin-driven anterograde transport of mitochondria [119,134]. Of note, because abnormal tau leads to microtubule disassembly [135], it is not excluded that impairing the axonal microtubule tracks for the transport of cargoes impacts mitochondrial transport. Conversely, knockdown of Milton or Miro that are adaptor proteins involved in the axonal transport of mitochondria enhanced tau phosphorylation in transgenic *Drosophila* expressing human tau in a process involving partitioning defective-1 (PAR-1) protein and leading to neurodegeneration [136]. This indicates that a loss of axonal mitochondria promotes tau phosphorylation and neuronal degeneration. In PC12 cells and cortical mouse neurons, abnormal tau was shown to inhibit mitochondrial movement in the neurite processes [121]. In this model, abnormal tau did not disturb the interaction between kinesin and microtubules, but caused an increase in the inter-microtubular distance, affecting mitochondrial movement and velocity. In line, a decrease in mitochondrial content in the neurites was quantified in neurons from rTg4510 mice expressing the P301L tau mutation, paralleled with a perinuclear clustering of mitochondria [120]. Similarly, a 50% reduction in the number of mitochondria was observed in the axons of P301L tau knock-in mice (KI-P301L), which express the tau mutation at physiological levels [122]. Interestingly, in this model, P301L tau was found to be hypophosphorylated, indicating that defects in axonal transport may not be due to tau abnormal hyperphosphorylation. However, KI-P301L mice presented an increased volume of motile axonal mitochondria as well as impairments in tau binding on microtubules, which may disturb mitochondrial transport. Similar observations were made in SH-SY5Y cells stably overexpressing the wild-type (hTau40) and mutant (P301L) form of tau [127]. Cells bearing the mutant form of tau presented a decreased mitochondrial movement, abnormal mitochondrial morphology (cristae with globular structures and branched membranes), a clustering of mitochondria around the nucleus, as well as decreased fusion/ fission rates compared to wild-type tau expressing cells.

Impairments in mitochondrial axonal transport were also evident in induced pluripotent stem cells (IPSCs) derived from frontotemporal dementia (FTD) patients bearing the R406W tau mutation [124]. In these IPSCs induced into cerebral organoids, axonal mitochondria were less stationary and moved more in the retrograde direction, resulting in fewer mitochondria into the axon when compared to control cells. In line, anterograde axonal transport of mitochondria was significantly reduced in IPSC-derived neurons bearing the N279K and P301L tau mutation compared to controls [123].

Taken together, these findings show that abnormal tau affects the axonal transport of mitochondria, decreasing the number of mitochondria at the synapse, which may lead to synaptic degeneration.

### 4.2. Mitochondrial Dynamics

Mutant tau protein was shown to impair mitochondrial dynamics in vivo in *Drosophila* expressing human R406W tau as well as rTg4510 and K3 mice, leading to the elongation of the mitochondrial network [125]. This may be involved in the reduced mitochondrial mobility and transport, as elongated mitochondria are not easily transported, especially along the axon. One proposed mechanism is that F-actin stabilization is increased in the presence of tau, disturbing the physical association of DRP1 and mitochondria, leading to DRP1 mislocalization, excessive mitochondrial elongation and subsequent neurotoxicity. These findings were reproduced in a recent study showing that increased levels of leucine-rich repeat kinase 2 (LRRK2), which is involved in PD, enhanced tau neurotoxicity by stabilizing the actin cytoskeleton, promoting DRP1 mislocalization and mitochondrial elongation [137]. Interestingly, mitochondrial elongation was already observed at the early stages of tau pathology in THY-Tau22 mice, when hippocampal Ca1 neurons are enriched with tau oligomers [138]. In these mice, DRP1 levels were significantly decreased at six months of age compared to age-matched wild-type littermates, whereas no differences were observed at 12 months of age. Another study demonstrated that human wild-type tau (htau) overexpression disrupts mitochondrial dynamics and causes mitochondrial elongation by increasing fusion proteins OPA1, and MFN1/ 2, which decreases neuronal viability [126]. MFN2 knockdown reduced the htau-enhanced mitochondrial fusion and restored mitochondrial function, indicating that mitofusin-associated mitochondrial fusion may play a role in tau toxicity.

Intriguingly, expression of caspase-cleaved tau in immortalized cortical neurons, as well as in cortical neurons from tau-/- knockout mice, induced mitochondrial fragmentation paralleled with a decrease of OPA1 levels [139]. This indicates that abnormal tau phosphorylation and tau truncation may impair mitochondrial dynamics via distinct mechanisms that still need to be unraveled.

### 4.3. Mitochondrial Bioenergetics

Tau-induced bioenergetic deficits were first observed in pR5 mice (P301L tau mutant mice), in which proteomic analysis revealed a downregulation of subunits of the mitochondrial Complexes I and V, together with an age-dependent decrease in mitochondrial respiration, Complex I activity and ATP levels, as well as an increase in ROS when compared to wild-type littermates [128,129]. These findings were recapitulated in vitro in SH-SY5Y cells overexpressing P301L tau (P301L cells) [127]. In addition, P301L cells presented a decrease in the maximal respiration and in the spare respiratory capacity, as well as a decrease in the ΔΨm when compared to wild-type tau overexpressing cells [130]. These data may be explained by the inhibition of Complex I activity induced by abnormal tau. In line, htau overexpression impaired mitochondrial bioenergetics by decreasing mitochondrial Complex I activity, ATP levels, as well as the ATP/ ADP ratio in HEK293 cells and hippocampus of htau mice compared to wild-type [126]. Interestingly, the genetic ablation of tau (tau-/-) significantly improved the bioenergetics capacity of mitochondria and reduced the oxidative damages in the hippocampus of young (three months old) mice, compared to age-matched wild-type littermates [132]. These improvements were paralleled with an increase in attentive capacity and exploratory ability in tau-/- mice, suggesting that preventing tau abnormal modifications enhances mitochondrial and brain functions.

A decrease in mitochondrial Complex I activity was also observed in the brain of rTg4510 mice, but was surprisingly paralleled with an increase in ΔΨm [140]. This feature was recapitulated in an advanced human neuronal model: IPSCs-derived neurons from FTDP-17 patients carrying the 10+16 mutation [141]. Compared to control IPSCs, FTDP-17 neurons presented a decrease in Complex I activity as well as in OXPHOS-derived ATP production, but an increased ΔΨm. One proposed mechanism is that hyperpolarization of mitochondria is due to Complex V working in reverse, leading to an increase in ROS production, oxidative stress and cell death.

Conversely, a recent report showed that tau decreases the ΔΨm via mitochondrial membrane poration, which compromised organelle structural integrity, leading to the swelling of mitochondria [142]. In this study, mitochondria were isolated from SH-SY5Y neuroblastoma cells and treated with tau oligomers. The decrease in ΔΨm was coupled with a release of cytochrome c. Intriguingly, the effects were independent of the mitochondrial permeability transition pore opening (mPTP), but rather due to the formation of non-selective ion-conducting tau nanopores caused by the binding of oligomeric tau on cardiolipin-rich membrane domains. These new data bring further insights into tau-induced mitochondrial toxicity.

### 4.4. Mitochondrial Permeability Transition Pore

The mPTP is a key contributor to cell death and has been involved in the pathophysiology of several neurodegenerative diseases [143]. Indeed, upon mPTP opening, mitochondrial membranes become permeable, disrupting mitochondrial function and releasing apoptotic signals into the cytosol. The exact composition of the mPTP remains elusive, but several reports suggested that it comprises proteins like voltage-dependent anion channel (VDAC) and translocator protein (TSPO) in the OMM, adenine nucleotide translocator (ANT) in the IMM, and cyclophilin D (CypD) in the mitochondrial matrix (reviewed in [143]). As abnormal tau was shown to disturb the ΔΨm in several models (see the previous section), one can suggest that these effects may be mediated by tau impacts on the mPTP.

Indeed, tau ablation inhibited mPTP formation in the hippocampus of three-months-old tau-/- mice by reducing the CypD protein level, compared to wild-type littermates [132]. Besides, tau was shown to directly interact with mitochondrial proteins, including subunits of the mitochondrial Complex V, which might explain bioenergetic deficits induced by abnormal tau, and VDAC [144]. In particular, phospho-tau interaction with VDAC was evident in the brain of AD patients at different Braak stages (I to V), as well as in 13-months-old APP/PS1 and 3xTgAD transgenic mice [145]. Furthermore, in our recent study, we showed that TSPO ligands increased the ΔΨm in htau- and P301L tau-overexpressing SH-SY5Y cells [131]. We speculated that this effect was mediated by the ability of these ligands to modulate the mPTP, although further investigations need to be conducted to determine the exact underlying mechanisms.

### 4.5. Mitophagy

Mitophagy plays a paramount role in mitochondrial quality control, by removing damaged mitochondria and ensuring a healthy mitochondrial population. In primary cultures of hippocampal neurons, the human 20–22 kDa NH2-tau fragment (NH2htau fragment mapping between 26 and 230 amino acids of the longest human tau isoform) was shown to increase mitophagic flux by recruiting Parkin to mitochondria, correlating with a decrease of synaptic stability [146,147], a feature also observed in human synaptic mitochondria from AD patients. In neurons expressing NH2htau, mitophagy inhibition partially prevented NH2htau-induced synaptic degeneration and neuronal death [147].

Other studies focusing on tau overexpression models showed that abnormal tau inhibits mitophagy [117,133]. Strikingly, an increase in the mitochondrial DNA/nuclear DNA ratio, as well as in mitophagy markers (COX IV and TOMM20), were observed in the brain of tau-positive AD patients, compared to tau-negative patients and healthy controls [133]. These data were recapitulated in vivo in htau transgenic mice, as well as in vitro in HEK293 and primary neurons overexpressing htau. In this study, htau overexpression induced an increase in the ΔΨm, preventing the recruitment of PINK/ Parkin to the mitochondrial fraction. Mitophagy deficits were rescued after Parkin overexpression in htau-overexpressing H293 cells. In line, using mitophagy reporters, Cummins and colleagues showed that both htau and P301L tau inhibited mitophagy in N2a cells and *Caenorhabditis elegans* [117]. Unlike the study previously described [137], the effects of tau on mitophagy were not due to changes in the ΔΨm, but to the sequestration of Parkin in the cytosol via interaction with the projection domain of tau. This sequestration prevented the recruitment of Parkin to mitochondria, inhibiting mitophagy. Interestingly, mitophagy stimulation reduced tau hyperphosphorylation in vitro (SH-SY5Y cells overexpressing 2N4R, 1N4R, 2N3R tau) and in vivo (transgenic nematodes expressing human tau and 3xTgAD mice), and reversed memory impairment in transgenic animals [86].

Together, these findings indicate that impaired mitophagy plays a role in tau pathogenesis, and highlight again distinct pathological features between models of tau phosphorylation and tau truncation.

## 5. New Insight on the Impact of Abnormal Tau on Neurosteroidogenesis and the ER-Mitochondria Coupling

### 5.1. Abnormal Tau and Neurosteroids

We recently showed that abnormal tau also disturbs another mitochondrial function: the synthesis of neurosteroids or neurosteroidogenesis [131]. Indeed, steroids can be synthesized de novo in the brain from cholesterol, independently of the peripheral steroidogenic glands, and are then called “neuro”-steroids (reviewed in [148]). The first step of steroidogenesis takes place in mitochondria with the transfer of cholesterol from the cytosol to the mitochondrial matrix, and its conversion to pregnenolone (PREG), the precursor of all neurosteroids. PREG is then converted into other neurosteroids either in mitochondria or in the ER. In the nervous system, neurosteroids play important roles in the regulation of neuronal functions as they can act as allosteric modulators of neurotransmitter receptors (e.g., NMDA or GABA receptors) [149].

Decreased levels of neurosteroids were observed in AD brains [150,151,152]. In particular, the levels of the neurosteroids pregnenolone sulfate (PREGS) and dehydroepiandrosterone sulfate (DHEAS) were significantly reduced in the striatum, hypothalamus and cerebellum of AD patients compared to non-demented controls (postmortem analysis) [150]. Lower levels of PREG, dehydroepiandrosterone (DHEA), as well as PREGS and DHEAS, were also observed in the hippocampus, amygdala and frontal cortex of AD patients, and were negatively correlated with the presence of NFTs. Interestingly, another study showed that the neurosteroid allopregnanolone is reduced in the prefrontal cortex, and is inversely correlated with the patient’s Braak stage, which reflects the evolution of tau pathology [151]. Together, these findings suggest a relationship between tau pathology, neurosteroids levels, and cognitive deficits, but the exact link remains elusive.

In our recent study, we showed that PREG levels were decreased in htau-overexpressing SH-SY5Y cells, and even more significantly reduced in P301L cells [131]. This effect was normalized in cells treated with TSPO ligands, which is involved in the first step of neurosteroidogenesis in mitochondria. The underlying mechanisms are currently under investigation in our laboratory. Nevertheless, we previously showed that neurosteroids, such as progesterone, estradiol, testosterone, DHEA and allopregnanolone, increase bioenergetics via the improvement of ATP production and mitochondrial respiration, and regulate the redox homeostasis in neuronal cells [153,154]. In particular, abnormal tau-induced mitochondrial impairments were reduced after treatment with progesterone, estradiol and testosterone [130].

Neuroprotective effects of a treatment with neurosteroids or sex-hormones-derived neuroactive steroids were evident against cognitive and bioenergetics deficits observed in AD (reviewed in [99,155]). In particular, allopregnanolone induces neurogenesis, restores learning and memory function, shows a trend to decrease phospho-tau levels, and reverses bioenergetic deficits in 3xTgAD transgenic mice [155,156,157,158]. Allopregnanolone is currently undergoing clinical trials for the treatment of AD [159].

As the effects of neurosteroids in other tauopathies are less studied and remain elusive, the use of these molecules as therapeutic agents against abnormal tau-induced neurodegeneration would deserve more attention in future investigations.

### 5.2. Abnormal Tau and ER-Mitochondria Coupling

Mitochondria are closely connected to ER membranes, forming a highly dynamic platform termed as mitochondria-associated ER membranes (MAMs). Particularly, up to 20% of the mitochondrial surface associates with ER membranes, allowing tight communication physically and biochemically. Accordingly, MAMs provide an outstanding scaffold for the crosstalk between mitochondria and the ER, playing a crucial role in different signaling pathways that require a rapid exchange of metabolites for the maintenance of cellular health. Moreover, numerous proteins have been revealed to be enriched in MAMs, participating in the regulation of many fundamental cellular pathways. Therefore, MAM crosstalk is involved in a variety of processes, including cholesterol metabolism, calcium homeostasis, phospholipid metabolism, the transfer of lipids between these two organelles, and the regulation of mitochondrial bioenergetics and dynamics. Besides, MAM coupling affects autophagy, ER-stress, inflammation and finally apoptosis [160,161,162,163]. In view of contributing to so many functions, it is hardly surprising that MAMs have drawn great attention in the studying of cell homeostasis and dysfunction, especially in the context of neurodegenerative disaeses. Strikingly, an increasing number of disease-associated proteins have been demonstrated to interact with MAMs, thereby regionally inducing structural and functional perturbations [164].

Mounting evidence emphasizes the role of a disturbed ER-mitochondria interconnection in neurodegenerative diseases such as AD, FTD and amyotrophic lateral sclerosis (ALS) (reviewed in [161]). For instance, in Aβ-related AD models, impairments in the ER-mitochondria coupling are translated by: (i) an increase in the expression of MAMs proteins and in the number ER-mitochondria contact points [165]; (ii) an upregulation in MAMs function including phospholipid and cholesterol synthesis [165,166]; and (iii) disturbed calcium homeostasis triggering a pathological cascade of events leading to apoptosis [160,167]. In amyotrophic lateral sclerosis with associated frontotemporal dementia (ALS/ FTD), TAR DNA-binding protein 43 (TDP-43) was shown to loosen ER–mitochondria contacts by disturbing the link between vesicle-associated membrane protein-associated protein B (VAPB) at the ER membrane and protein tyrosine phosphatase interacting protein 51 (PTPIP51) at the mitochondrial membrane, two proteins involved in MAMs tethering [161,168]. This disruption of the ER-mitochondria interaction disturbed the calcium exchange between both organelles, and may be linked to the decrease in ATP levels leading to motor neuron degeneration [168,169].

Until recently, only one study had focused on the impact of abnormal tau on the ER-mitochondria interaction [170]. Indeed, using electron microscopy techniques, Perreault and colleagues showed a higher number of ER-mitochondria contact points in a tau transgenic mouse model (JNPL3, P301L tau transgenic mice) compared to wild-type animals. An increased association of hyperphosphorylated tau with ER membranes was also observed in post-mortem brains of AD patients, suggesting that the ER-mitochondria axis may also play a role in abnormal tau-induced neurodegeneration. In line, Cieri et al. showed that overexpression of caspase 3-cleaved 2N4R∆C_20_ tau in Hela cells increased the number of tight (8–10 nm) ER-mitochondria contact points, whereas long-range (40–50 nm) interactions were not affected [171]. In parallel, truncated (2N4R∆C_20_) and full length (2N4R) tau expression affected the ER calcium storage, suggesting that tau may disturb the MAMs leading to ER calcium mishandling. Interestingly, tau was also found at the OMM and within the IMS, but not in the mitochondrial matrix.

With the aim to characterize the link between ER-stress and bioenergetic defects in the presence of tau, we recently showed that P301L expressing SH-SY5Y cells presented a highly activated unfolded protein response (UPR = ER-stress) and dysregulated mitochondrial bioenergetics in basal condition, when compared to wtTau cells [172]. Furthermore, acute ER-stress was induced using thapsigargin, which increased the activity of the UPR in both wtTau and P301L tau cells, leading to the upregulation of apoptotic pathways, and further dysregulated mitochondrial function. This study supports the role of close communication between mitochondria and the ER during apoptosis in tauopathies.

Further investigations are now needed to unravel the underlying mechanisms, as well as potential effects of abnormal tau on other MAM functions (e.g., cholesterol and phospholipid homeostasis), which may highlight potential therapeutic targets. For instance, in ALS/ FTD, TDP-43 was shown to induce the activation of GSK 3β, which then disrupts the binding of PTPIP51 and VAPB (reviewed in [161]). Since GSK 3β is also involved in tau phosphorylation and is up-regulated in AD [173,174], it constitutes a good candidate against tauopathies. However, whether GSK 3β is involved in impairments of the ER-mitochondria coupling in abnormal tau-related diseases remains to be determined.

## 6. Conclusions

In summary, abnormal tau has an impact on all aspects of mitochondrial functions, from mitochondrial transport and dynamics, to bioenergetics and mitophagy (Figure 3 and Figure 4). Because mitochondria play a pivotal role in neuron life and death [1], alteration in their function leads to neuronal dysfunction and death, and eventually to dementia. Besides, given that mitochondria are not isolated, self-autonomous organelles floating in the cytosol, but on the contrary highly interconnected with other cellular compartments, including the ER, it is not surprising that abnormal tau-induced mitochondrial dysfunction affects other cellular functions, and vice versa. It is, however, important to note that discrepancies are sometimes observed between tau models. Common features were observed with regards to mitochondrial bioenergetics (decreased ATP levels, Complex I inhibition) and transport (impaired axonal transport, perinuclear clustering), but different findings were obtained concerning the ΔΨm (increased or decreased ΔΨm in the presence of tau), mitochondrial dynamics (tau-induced mitochondrial elongation or fragmentation) and mitophagy (triggered or inhibited by tau). Strikingly, these differences are often linked with the type of tau used in the study (phospho-tau or truncated tau). These two aspects of tau pathology should be considered to fully understand the molecular mechanisms underlying tauopathies, and are therefore important to highlight therapeutic targets.

## Figures and Tables

**Figure 1 ijms-21-06344-f001:**
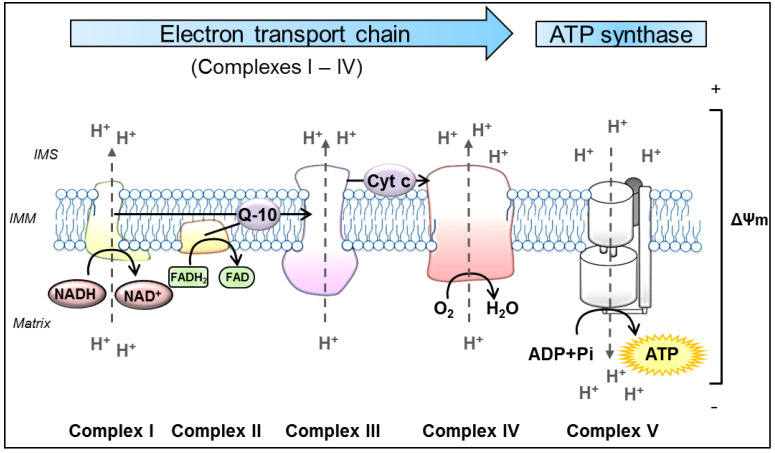
Schematic representation of oxidative phosphorylation (OXPHOS) in mitochondria. The process of OXPHOS is the main pathway in the cell to produce adenosine triphosphate (ATP). It consists of two coupled processes embedded in the inner mitochondrial membrane (IMM), the electron transport chain (ETC) and the ATP synthesis. In the ETC, the reduced substrates NADH and FADH_2_ are oxidized by the NADH-ubiquinone oxidoreductase (Complex I) and succinate-CoenzymeQ reductase (Complex II), respectively. These proteins transfer electrons from their substrates onto Q-10, which serves as a substrate for the CoenzymeQ-cytochrome c oxidoreductase (Complex III). Q-10 is a highly lipophilic substance that is able to diffuse within the IMM. Complex III transfers the electrons from Q-10 onto cytochrome c, which is a water-soluble electron carrier, located at the surface of the IMM in the IMS. In the final step of the ETC, cytochrome c oxidase (Complex IV) uses the electrons from reduced Cyt c to reduce molecular oxygen to water. In the process of transferring electrons, the Complexes I, III and IV actively move protons (H^+^) from the mitochondrial matrix to the IMS, forming the ΔΨm. Ultimately, this potential is used by the ATP synthase (Complex V) to catalyze the generation of ATP from ADP and Pi. ADP: adenosine diphosphate, ATP: adenosine triphosphate, Cyt c: cytochrome c, ETC: electron transport chain, FADH2: flavin adenine dinucleotide, IMM: inner mitochondrial membrane, IMS: intermembrane space, NADH: nicotinamide adenine dinucleotide, Pi: inorganic phosphate, Q-10: coenzymeQ10, ΔΨm: mitochondrial membrane potential.

**Figure 2 ijms-21-06344-f002:**
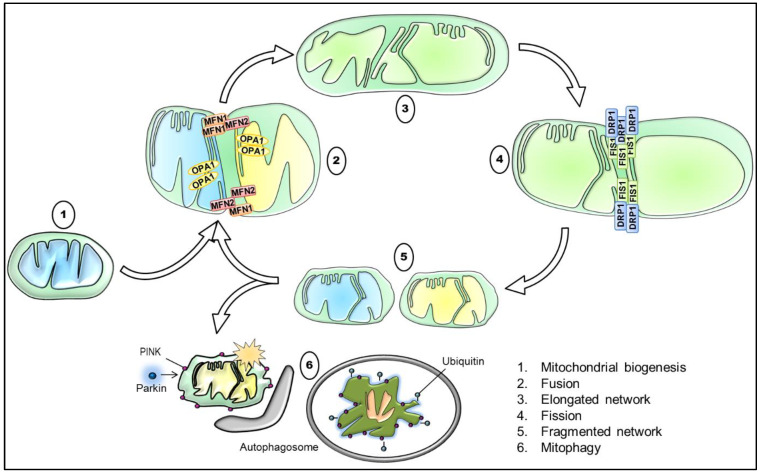
Schematic illustration of the interplay between mitochondrial biogenesis, fusion, fission, and mitophagy with key proteins involved. Briefly, mitochondrial biogenesis generates functional mitochondria, for instance in response to a reduced mitochondrial mass. The tethering of two mitochondria via the outer mitochondrial membrane (OMM) and IMM mediated through MFN1/ MFN2 and OPA1, respectively, results in their fusion, and thus in the elongation of the mitochondrial network. Contrarily, the recruitment and orchestration of primarily DRP1 and assisting proteins, such as FIS1, causes mitochondrial division. Consequently, the process of mitochondrial fission promotes a more fragmented mitochondrial network and is required for the removal of damaged and dysfunctional mitochondria. Lastly, the accumulation of PINK1 and the subsequent recruitment of Parkin target defective mitochondria that are subsequently degraded by mitophagy. DRP1: dynamin-related protein 1, FIS1: mitochondrial fission protein 1, IMM: inner mitochondrial membrane, MFN1: mitofusin 1, MFN2: mitofusin 2, OMM: outer mitochondrial membrane, OPA1: optic atrophy 1, PINK1: PTEN-induced kinase 1.

**Figure 3 ijms-21-06344-f003:**
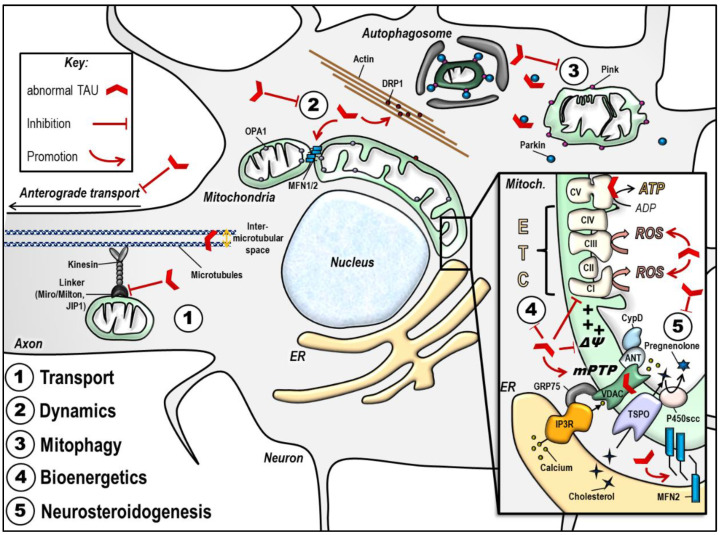
Abnormal tau protein impairs mitochondrial function. The scheme summarizes the impact of disease-associated tau protein on the different aspects of mitochondrial function (see details in the text). Of note, the effects illustrated here may be different according to tau models used (phospho-tau versus truncated tau). (**1**) Abnormal tau inhibits anterograde transport of mitochondria along the axon, leading to a decreased number of mitochondria at the synapse, and mitochondrial perinuclear clustering. (**2**) Abnormal tau seems to trigger mitochondrial network elongation by increasing MFN2 levels and by triggering DRP1 mislocalization and clustering in actin filaments. (**3**) Abnormal tau inhibits mitophagy by interacting with Parkin. (**4**) Abnormal tau disturbs mitochondrial bioenergetics by inhibiting Complex I activity, decreasing the ΔΨm and ATP levels, and increasing ROS production. These effects may also be linked to abnormal tau impacts on the mPTP opening, and/ or to tau direct binding on mitochondrial proteins like voltage-dependent anion channel (VDAC) and subunits on the respiratory Complex V. (**5**) Abnormal tau disturbs mitochondrial steroidogenesis by decreasing pregnenolone synthesis. Finally, abnormal tau seems to impact on the ER-mitochondrial coupling, which may have consequences on all the above mentioned mitochondrial functions. ANT: adenine nucleotide translocator, ATP: adenosine triphosphate, CI-CV: respiratory complexes I–V, CypD: cyclophilin D, DRP1: dynamin-related protein 1, ER: endoplasmic reticulum, ETC: electron transport chain, GRP75: glucose-related protein 75, IP3R: inositol 3 phosphate receptor, JIP1: c-Jun N-terminal kinase-interacting protein 1, MFN1/ 2: mitofusin 1/ 2, mPTP: mitochondrial permeability transition pore, OPA1: optic atrophy 1, P450scc: cytochrome P450 cholesterol side-chain cleavage enzyme, ROS: reactive oxygen species, TSPO: translocator protein, VDAC: voltage-dependent anion channel, ΔΨm: mitochondrial membrane potential.

**Figure 4 ijms-21-06344-f004:**
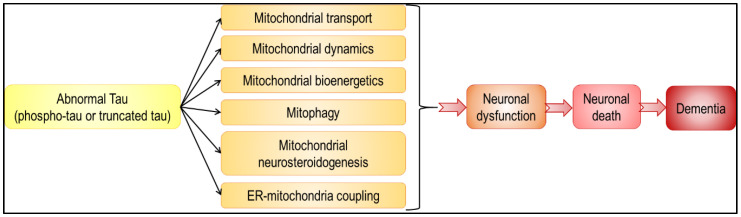
Abnormal tau-induced mitochondrial impairments may lead to neuronal death and dementia. Abnormal tau was shown to have a negative impact on all aspects of mitochondrial function (black arrows). These tau-induced disturbances may lead to various neuronal dysfunctions, ranging from subtle alterations in neuronal physiology to cell death and neurodegeneration (red arrows).

**Table 1 ijms-21-06344-t001:** Impacts of tau on mitochondrial function in vitro and in vivo.

Mitochondrial Function	Model	Main Impairments in the Presence of Abnormal Tau Versus Respective Controls	Reference
**Transport**	K3 mice (human K369I mutant tau)	Impairment in anterograde (not retrograde) transport of mitochondria along the axon	[119]
	rTg4510 mice (repressible human P301L mutant tau)	Decreased percentage of the cytoplasm occupied by mitochondria Reduction of mitochondrial content in neurites Perinuclear clustering of mitochondria with no change in mitochondrial volume	[120]
	PC12 cells and cortical neurons expressing tau mutants: 3A (non-phosphorylatable) and 3D (phosphorylation mimic), with mutations at the AT8 sites (S199, S202, and T205)	Increase in stationary mitochondria, decrease in the velocity of mitochondrial movement Increase in the inter-microtubular spacing affecting mitochondrial movement	[121]
	KI-P301L mice (P301L tau knock-in)	Reduced number of axonal mitochondria Increased volume of motile mitochondria in the axons Impaired binding of tau to microtubules	[122]
	IPSC-derived neurons with tau mutations	Reduced number of axonal mitochondria and increase retrograde transport (IPSCs with R406W tau mutation) Decreased anterograde transport (IPSCs with N279K and P301L tau mutations	[123,124]
**Dynamics**	*Drosophila* expressing human wild-type tau or human R406W mutant tau rTg4510 and K3 mice	Excessive mitochondrial elongation Increased actin stabilization and decreased localization of dynamin-related protein 1 (DRP1) to mitochondria	[125]
	HEK293 cells and rat primary hippocampal neurons expressing the human wild-type full-length tau (hTau) hTau mice (STOCK Mapttm1(EGFP)Klt Tg(MAPT)8cPdav/J)	Disruption of mitochondrial dynamics, enhanced fusion and perinuclear accumulation of mitochondria Increased expression of fusion proteins mitofusin 1 (MFN1), mitofusin 2 (MFN2) and optic atrophy 1 (OPA1), reduced ubiquitination of MFN2	[126]
	SH-SY5Y cells stably overexpressing wild-type (wt) and P301L mutant tau	Changes in mitochondrial morphology, decreased fusion and fission rates Clustering of mitochondria around the nucleus and decreased mitochondrial movement	[127]
**Bioenergetics**	pR5 mice (human P301L mutant tau)	Decreased mitochondrial respiration, mitochondrial Complex I activity, adenosine triphosphate (ATP) levels Increased reactive oxygen species (ROS) levels and superoxide anion radicals (O_2_^•−^)	[128,129]
	SH-SY5Y cells stably overexpressing wild-type (wt) and P301L mutant tau	Decreased mitochondrial respiration, mitochondrial Complex I activity, ATP levels, and mitochondrial membrane potential (ΔΨm)	[127,130,131]
	HEK293 cells expressing the human wild-type full-length tau (hTau) hTau mice	Decreased mitochondrial Complex I activity, ATP levels, and ATP/ ADP ratio	[126]
**mPTP**	Three-months-old tau knockout (tau-/-) mice	Inhibition of mitochondrial permeability transition pore (mPTP) formation in the hippocampus, reduction of cyclophilin D (CypD) protein level	[132]
**Mitophagy**	AD patients hTau mice HEK293 expressing hTau	Increase of mitophagy markers (COX IV, TOMM20, ratio mtDNA/ nDNA) Dose-dependent allocation of tau proteins into the outer mitochondrial membrane (OMM) Increased ΔΨm, which impairs the mitochondrial residence of PTEN-induced kinase 1 (PINK1)/ Parkin	[133]
	N2a cells and *Caenorhabditis elegans* expressing human wild-type (hTau) and P301L mutant tau	Decreased mitophagy Sequestration of Parkin in the cytosol, preventing its recruitment to defective mitochondria	[117]
**Neuro-** **steroidogenesis**	SH-SY5Y cells stably overexpressing wild-type (wt) and P301L mutant tau	Decreased pregnenolone synthesis	[131]

AD: Alzheimer’s disease, COX IV: cytochrome c oxidase subunit IV, FTD: frontotemporal dementia, IPSCs: induced pluripotent stem cells, TOMM20: translocase of outer mitochondrial membrane 20, mtDNA/ nDNA: mitochondrial DNA/ nuclear DNA.

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
