# Peer review of "Insights into Disease-Associated Tau Impact on Mitochondria"

_ijms, 2020, doi:10.3390/ijms21176344_

Round 1
Reviewer 1 Report
IJMS-878500 // New insights into disease-associated tau impact on mitochondria.
The authors provide a comprehensive yet concise review of tau and mitochondria in relation to neurodegeneration before going on to describe the evidence that tau impairs mitochondrial function in several aspects. These are nicely summarised in Table 1 and drawn schematically in Figure 3. This is a timely review since it highlights the need for further investigation in this area.
The review puts together, in a single place, the background to discuss the tau-mitochondrial interconnections that may be involved in Alzheimer’s and other neurodegenerative tauopathies. It would be worth highlighting that nearly all of the data is derived from transgenic mouse or in vitro studies. Table 3 only lists one example [98] where AD patients have been used as the basis for their thesis. Whilst it may fit the thesis that they provide a “mitocentric” picture of tau toxicity, I am not sure that this is best put in this way. It may be that there are several pathways that lead to neurodegeneration, and not all may be through mitochondria. Although the authors state in the abstract that mitochondrial dysfunction arises as an early event that occurs before cognitive decline, this also applies to tau aggregation as well as amyloid deposition. I think that the data is in need of further investigation, as suggested by the authors themselves.
The authors seem to equate, like many others, that an extensive literature on tau phosphorylation leads to a “well-established role of phosphorylation in tau pathology”. However wording that precedes that quote (line 124) sounds less certain (“assumed” on line 167) that it is the major driving factor behind tau pathology. There is no definitive proof that phosphorylation leads to PHF assembly.
Line 295: “Mitochondrial shape is fundamental for …activity” There are many different shapes of mitochondria, even under healthy states – would it not be better to refer to the “integrity” of the mitochondria rather than their shape?
In mitophagy section, it would be helpful to describe the amino-truncated tau fragment as mapping to within residues 26-230 [130].
Minor points
There are occasional instances of language requiring attention, e.g.:
Abst, line 19 - ..in detail the..
Line 283: …and the subsequent recruitmaent… (change from “following”)
Line 520: “less know” should be “less well known”, or better still omitted altogether.
Table 1: Drosophila and C elegans in italic.
MAPT – should be italic when referring to the human tau gene; and there is no need to use MAPT as an abbreviation for the protein, when tau is abbreviated even more.
In Fig 2, there is “Pink” instead of PINK.
Line 554: “Interestingly, fraction of tau was…” is grammatically incorrect but should have used “fragment”. Thus: “interestingly, a fragment of tau was…”
Line 578-581: “…important to highlight therapeutic targets.” This last sentence should be better written and, perhaps, split into two.
Citations [86] and [105] are the same.
Author Response
The authors provide a comprehensive yet concise review of tau and mitochondria in relation to neurodegeneration before going on to describe the evidence that tau impairs mitochondrial function in several aspects. These are nicely summarised in Table 1 and drawn schematically in Figure 3. This is a timely review since it highlights the need for further investigation in this area.
The review puts together, in a single place, the background to discuss the tau-mitochondrial interconnections that may be involved in Alzheimer’s and other neurodegenerative tauopathies. It would be worth highlighting that nearly all of the data is derived from transgenic mouse or in vitro studies. Table 3 only lists one example [98] where AD patients have been used as the basis for their thesis. Whilst it may fit the thesis that they provide a “mitocentric” picture of tau toxicity, I am not sure that this is best put in this way. It may be that there are several pathways that lead to neurodegeneration, and not all may be through mitochondria.
Answer: We thank the reviewer for highlighting this important point. By using the term “mitocentric”, our idea was to put in evidence all the aspects of mitochondrial physiology that are impacted by the abnormal tau protein. This does not exclude, of course, that tau has many other effects on other cellular functions that lead to neuronal death. We therefore add the following statements in the text. Please see:
- Lines 53-55: “Besides, abnormal tau has many effects on other cellular functions that may lead to neurodegeneration, which are nicely reviewed elsewhere [8-10].”
It is also true that nearly all the data discussed in the Section 4, and presented in Table 1, come from in vivo and in vitro study. This important point was highlighted for the reader, please see:
- Lines 414-417: “Noteworthy, nearly all the data discussed here derived from in vitro and animal studies. Therefore, studies performed on patients with tauopathies are highly needed, in order to confirm and fully apprehend mitochondrial dysfunctions induced by abnormal tau protein.”
Although the authors state in the abstract that mitochondrial dysfunction arises as an early event that occurs before cognitive decline, this also applies to tau aggregation as well as amyloid deposition. I think that the data is in need of further investigation, as suggested by the authors themselves.
Answer: We thank the reviewer for this comment. Mitochondrial dysfunction was indeed observed in AD before cognitive deficits, Aβ and tau pathology, and previously discussed elsewhere (see Schmitt K., et al, ARS 2012). Further investigation are highly required, especially human studies focused exclusively on “pure” tauopathies, without amyloid component.
The authors seem to equate, like many others, that an extensive literature on tau phosphorylation leads to a “well-established role of phosphorylation in tau pathology”. However wording that precedes that quote (line 124) sounds less certain (“assumed” on line 167) that it is the major driving factor behind tau pathology. There is no definitive proof that phosphorylation leads to PHF assembly.
Answer: We thank the reviewer for this comment. We agree that the role of abnormal tau phosphorylation in PHF assembly still remains to be proven, and that other tau post-translational modification (e.g. tau truncation) may be involved. Therefore, we changed the sentence accordingly. Please see lines 133-134: “Nevertheless, in contrast to the more investigated role of phosphorylation in tau pathology, the implication of the other tau post-translational modifications is yet to be fully characterized.”
Line 295: “Mitochondrial shape is fundamental for …activity” There are many different shapes of mitochondria, even under healthy states – would it not be better to refer to the “integrity” of the mitochondria rather than their shape?
Answer: We agree. The term “mitochondrial shape” was replaced with “mitochondrial integrity”. Please see line 346.
In mitophagy section, it would be helpful to describe the amino-truncated tau fragment as mapping to within residues 26-230 [130].
Answer: We agree with this comment. Further details were added. Please see lines 545-547: “In primary cultures of hippocampal neurons, the human 20-22 kDa NH2-tau fragment (NH2htau fragment mapping between 26 and 230 amino acids of the longest human tau isoform) was shown to increase mitophagic flux by recruiting Parkin to mitochondria,…”
Minor points
There are occasional instances of language requiring attention, e.g.:
Abst, line 19 - ..in detail the..
Line 283: …and the subsequent recruitmaent… (change from “following”)
Line 520: “less know” should be “less well known”, or better still omitted altogether.
Table 1: Drosophila and C elegans in italic.
MAPT – should be italic when referring to the human tau gene; and there is no need to use MAPT as an abbreviation for the protein, when tau is abbreviated even more.
In Fig 2, there is “Pink” instead of PINK.
Line 554: “Interestingly, fraction of tau was…” is grammatically incorrect but should have used “fragment”. Thus: “interestingly, a fragment of tau was…”
Line 578-581: “…important to highlight therapeutic targets.” This last sentence should be better written and, perhaps, split into two.
Citations [86] and [105] are the same.
Answer: We thank the reviewer for making us aware of these mistakes. All the above mentioned points were corrected in the text, table and figures.
Reviewer 2 Report
The review article by Szabo et al., titled as “New insights into disease-associated Tau impact on mitochondria” contains interesting perspectives of tauopathies in neuroinflammatory disorders. Authors have well emphasized the role of disease-associated tau and mitochondrial dysfunction in neurodegenerative disorder. However, I have few concerns which are listed below for authors to improve this manuscript –
Major Concerns-
- As the title suggests “new insights”, mentioned in abstract, are neurosteroidogenesis and mitochondrial-ER coupling. Surprisingly these mechanisms are least discussed in this article, specially neurosteroidogenesis. To justify the title, authors needs to provide more details of neurosteroidogenesis and speculate its interconnection with neurodegenerative disorders (primary and secondary tauopathies).
- Authors are suggested to include more details to justify why these new targets can be approached for therapeutic interventions.
- Authors are suggested to include the relevant highlights which should justify the title.
Minor Concerns-
- In figure 1, to establishes an electrochemical gradient normally complex I, III, and IV are involved in pumping H+ out of the matrix and into the intermembrane space, not from intermembrane space to the matrix. Authors have used double head arrows to show the flow of H+ instead using single head arrows. Please justify this.
Trivial suggestions
- In figure 2, last step (mitophagy) is 6th while it is mentioned as 5th. Authors need to correct the typo error
Author Response
As the title suggests “new insights”, mentioned in abstract, are neurosteroidogenesis and mitochondrial-ER coupling. Surprisingly these mechanisms are least discussed in this article, specially neurosteroidogenesis. To justify the title, authors needs to provide more details of neurosteroidogenesis and speculate its interconnection with neurodegenerative disorders (primary and secondary tauopathies).
Authors are suggested to include more details to justify why these new targets can be approached for therapeutic interventions.
Authors are suggested to include the relevant highlights which should justify the title.
Answer: We thank the reviewer for these constructive comments. For the sake of clarity, the work “new” was removed from the title. The paragraph about neurosteroidogenesis was moved in the last section, which is now titled: “New Insight on the Impact of Abnormal Tau on Neurosteroidogenesis and the ER-Mitochondria Coupling”. Further details about neurosteroidogenesis, their implication of neurosteroids in AD, as well as their therapeutic potential, were added. Please see:
- Lines 597-633: “We recently showed that abnormal tau also disturbs another mitochondrial function: the synthesis of neurosteroids or neurosteroidogenesis [131]. Indeed, steroids can be synthesized de novo in the brain from cholesterol, independently of the peripheral steroidogenic glands, and are then called “neuro”-steroids (reviewed in [149]). The first step of steroidogenesis takes place in mitochondria with the transfer of cholesterol from the cytosol to the mitochondrial matrix, and its conversion to pregnenolone (PREG), precursor of all neurosteroids. PREG is then converted into other neurosteroids either in mitochondria or in the ER. In the nervous system, neurosteroids play important roles in the regulation of neuronal functions as they can act as allosteric modulators of neurotransmitter receptors (e.g. NMDA or GABA receptors) [150]. In our recent study, we showed that PREG levels were decreased in htau-overexpressing SH-SY5Y cells, and even more significantly reduced in P301L cells [131]. This effect was normalized in cells treated with TSPO ligands, which is involved in the first step of neurosteroidogenesis in mitochondria. The underlying mechanisms are currently under investigation in our laboratory. Nevertheless, we previously showed that neurosteroids, such as progesterone, estradiol, testosterone, DHEA and allopregnanolone, increase bioenergetics via the improvement of ATP production and mitochondrial respiration, and regulate the redox homeostasis in neuronal cells [154,155]. In particular, abnormal tau-induced mitochondrial impairments were reduced after treatment with progesterone, estradiol and testosterone [130]. As the effects of neurosteroids in other tauopathies are less studied and remain elusive, the use of these molecules as therapeutic agents against abnormal tau-induced neurodegeneration would deserve more attention in future investigations.”
- Neuroprotective effects of a treatment with neurosteroids or sex-hormones-derived neuroactive steroids were evident against cognitive and bioenergetics deficits observed in AD (reviewed in [156,157]). In particular, allopregnanolone induces neurogenesis, restores learning and memory function, shows a trend to decrease phosphor-tau levels, and reverses bioenergetic deficits in 3xTgAD transgenic mice [156,158-160]. Allopregnanolone is currently undergoing clinical trials for the treatment of AD [161].
- Decreased levels of neurosteroids were observed in AD brains [151-153]. In particular, the levels of the neurosteroids pregnenolone sulfate (PREGS) and dehydroepiandrosterone sulfate (DHEAS) were significantly reduced in the striatum, hypothalamus and cerebellum of AD patients compared to non-demented controls (postmortem analysis) [151]. Lower levels of PREG, dehydroepiandrosterone (DHEA), as well as PREGS and DHEAS, were also observed in the hippocampus, amygdala and frontal cortex of AD patients, and were negatively correlated with the presence of NFTs. Interestingly, another study showed that the neurosteroid allopregnanolone is reduced in prefrontal cortex, and is inversely correlated with the patient’s Braak stage, which reflects the evolution of tau pathology [152]. Together, these findings suggest a relationship between tau pathology, neurosteroids levels, and cognitive deficits, but the exact link remains elusive.
We also bring further examples about impairments in the ER-mitochondria coupling in neurodegenerative diseases. Please see:
- Lines 656-663: “In amyotrophic lateral sclerosis with associated frontotemporal dementia (ALS/ FTD), TAR DNA-binding protein 43 (TDP-43) was shown to loosen ER–mitochondria contacts by disturbing the link between VAPB (vesicle-associated membrane protein-associated protein B) at the ER membrane and PTPIP51 (protein tyrosine phosphatase interacting protein 51) at the mitochondrial membrane, two proteins involved in MAMs tethering [163,170]. This disruption of the ER-mitochondria interaction disturbed the calcium exchange between both organelles, and may be linked to the decrease in ATP levels leading to motor neuron degeneration [170,171].
- Lines 684-691: “Further investigations are now needed to unravel the underlying mechanisms, as well as potential effects of abnormal tau on other MAM functions (e.g. cholesterol and phospholipid homeostasis), which may highlight potential therapeutic targets. For instance, in ALS/ FTD, TDP-43 was shown to induce the activation of GSK 3β, which then disrupts the binding of PTPIP51 and VAPB (reviewed in [163]). Since GSK 3β is also involved in tau phosphorylation and is up-regulated in AD [175,176], it constitutes a good candidate against tauopathies. However, whether GSK 3β is involved in impairments of the ER-mitochondria coupling in abnormal tau-related diseases remains to be determined.”
Minor Concerns-
In figure 1, to establishes an electrochemical gradient normally complex I, III, and IV are involved in pumping H+ out of the matrix and into the intermembrane space, not from intermembrane space to the matrix. Authors have used double head arrows to show the flow of H+ instead using single head arrows. Please justify this.
Answer: We are grateful to reviewer for making us aware of this point. The double head arrows were a mistake, which is now corrected. Please see Figure 1.
Trivial suggestions
In figure 2, last step (mitophagy) is 6th while it is mentioned as 5th. Authors need to correct the typo error
Answer: We thank the reviewer for making us aware of this typo error, which is now corrected. Please see Figure 2.
Reviewer 3 Report
In this interesting review, the authors discuss in details the different impacts of neurodegenerative disease-associated tau protein on mitochondrial functions, according many aspects for the subjects. As reviewer, the subject is very interesting and extremely important for the understanding of tauopathies involving mitochondrial dysfunction. The writing of the manuscript is well done, the subject is well documented (116 cited references!) with relevant examples (both in molecular, biological and physiological domains) and the work of the authors is well integrated. This synthesis work certainly deserves to be published in the journal IJMS. However, many points concerning notably the clarity of the manuscript detract from the quality of this review. Thus, these preclude accepting immediately the article for publication in IJMS.
1/ in the section INTRODUCTION, the objectives of the review are well explained and we understand the whole issue of the study. However, the authors should review the order of the points to better understand the relationship between tau and mitochondrial dysfunctions in a context of tauopathy: as a piece of advice, it might be better to reverse paragraphs 1 and 2 of this introductory part quite simply.
2/ - p2 lines 87-90: it is important that the authors specify that the proline-rich region (in particular with the PXXP motifs) actively contributes to the interaction of the Tau protein with the microtubules (see all the work of the teams of Feinstein or Mandelkow for eg).
3/ - p2-3 lines 91-98: the authors should add a few words on the role of cysteines in the sequence of tau, namely in the microtubule-binding domain (2 cys for the 4R isoform and 1 for 3R). The redox processes of these cysteines appear to contribute to the pathogenicity of the protein
4/ - p3 lines 134-135: I agree with the authors when they write that "binding of tau to microtubules is a highly dynamic process" but the dynamicity of this tau-MT binding is not ONLY regulated by phosphorylation. The tau-MT interaction process is fundamentally dynamic, independent of any post-transcriptional modification of tau, and in particular defined by a biochemical exchange of tau with the MT surface and which can be measured / defined by kinetic constants Ka and Kd . The authors surely want to insist here on the fact that the phosphorylation of tau (and particularly on MTBD! A part to be added / clarified by the authors, I think) can drastically modify this biochemical exchange. It is therefore crucial to rewrite this part, supported by examples.
5/ - p4 lines 168-182: the authors here describe the involvement of tau hyperphosphorylation in tauopathies. However, with nearly 80 phosphorylabl sites, it is important to show and explain which ones are extremely important for the "loss of normal microtubule-stabilizing fraction" (l172), and in particular which sites are important for the generation of PHFs / NFTs.
6/ - p5-6 lines 210-264: descriptions of the functioning of the mitochondria and its role as a factory to produce energy are obviously necessary for the novice, and the authors have carried out a very advanced editorial work. Perhaps a little too precisely, in particular in the passage devoted to the mechanisms of production of oxygenated species in which we do not see very well where the authors are coming from. We have to wait for the last sentence (lines 262-264) to understand that this whole description is important in neurodegenerative diseases (by the way, really all? Probably not, and only concerns those for which a dysfunction of the mitochondria is proven!) . Consequently, the reader remains a little unsatisfied since the authors do not describe the physiopathological studies illustrating their remarks. It would therefore be desirable to rewrite this part and rely on examples of studies.
7/ - p6-9 lines 265-358: the description of the dynamics of mitochondria approached both at the protein and physiological level is extremely interesting and I think complete. However, the authors should consider this description in neurophysiopathological context: what are the consequences of ROS on the formation of PHF / NFT for example? Once again, this is not illustrated enough of a concrete example (e.g. studies carried out on cell lines, in vivo, patient cohort ...) and it is only towards the end of this part that we learns all the interest of the description in neurodegenerative diseases.
8/ - p6 line 360-374: what is the interest of this section in the tau-mitochondria subject discussed here?
9/ - p12-13: Table 1 presenting the points implicating the tau protein in the dysfunction of the mitochondria is very useful. The authors also describe some of the most important points. Suddenly, this table could perhaps be moved before the authors' text, indicating the important points that will be taken up by the authors.
10/ - p13 paragraph “5. New insight ...”: I finally understand why the authors wrote above a part devoted to the ER-mitochondria coupling. In view of the short length of the previous paragraph (p9 lines 359-374), the authors could move it here as an introduction to this part 5.
11/ - p15: where is the comment for the figure 3 in the manuscript? In addition, this schema which tries to summarize the different targets of tau in the 5 processes involving the mitochondria (transport, dynamics, etc.) is quite difficult to understand: the authors should specify point by point whether tau has a promoting or disrupting role (maybe with another symbol for tau in the picture?) on the illustrated regulation (directly in the schema, and not only in the legend of figure). Also, it is difficult to understand from this rather complex schema the relationship between tau protein, the dysfunction of mitochondria and the appearance of the neurodegenerative disease, what about the effect of the hyperphosphorylation of tau, of possible mutations, of the ratio of expression between the 3R and 4R isoforms, etc ... in the disease? It is indeed difficult to synthesize all the points raised in the review by such a schema.
Author Response
1/ in the section INTRODUCTION, the objectives of the review are well explained and we understand the whole issue of the study. However, the authors should review the order of the points to better understand the relationship between tau and mitochondrial dysfunctions in a context of tauopathy: as a piece of advice, it might be better to reverse paragraphs 1 and 2 of this introductory part quite simply.
Answer: We thank the reviewer for this comment. Paragraph 1 and 2 of the INTRODUCTION are now inverted. Please see lines 29-51.
2/ - p2 lines 87-90: it is important that the authors specify that the proline-rich region (in particular with the PXXP motifs) actively contributes to the interaction of the Tau protein with the microtubules (see all the work of the teams of Feinstein or Mandelkow for eg).
Answer: We agree with the referee. Please see the changes made lines 94-98: “Furthermore, the ability of tau to interact with microtubules is mediated by the microtubule-binding domain in combination with the adjacent proline-rich flanking domains. Whereas the microtubule-binding repeats bind only weakly to microtubules (but possess specificity for microtubule assembly), the proline-rich region provides an efficient targeting to the microtubule surface].”
3/ - p2-3 lines 91-98: the authors should add a few words on the role of cysteines in the sequence of tau, namely in the microtubule-binding domain (2 cys for the 4R isoform and 1 for 3R). The redox processes of these cysteines appear to contribute to the pathogenicity of the protein
Answer: We agree. The following information were added. Please see lines 82-85: “Besides, depending on the isoform tau contains either one or two cysteine residues in the microtubule-binding domain. While in the 3R isoform only C322 within the third repeat is present, 4R tau additionally comprises C291 within the fourth repeat. This variance seems to have an influence on the assembly of paired helical filaments (PHFs) in vitro [21].”
4/ - p3 lines 134-135: I agree with the authors when they write that "binding of tau to microtubules is a highly dynamic process" but the dynamicity of this tau-MT binding is not ONLY regulated by phosphorylation. The tau-MT interaction process is fundamentally dynamic, independent of any post-transcriptional modification of tau, and in particular defined by a biochemical exchange of tau with the MT surface and which can be measured / defined by kinetic constants Ka and Kd . The authors surely want to insist here on the fact that the phosphorylation of tau (and particularly on MTBD! A part to be added / clarified by the authors, I think) can drastically modify this biochemical exchange. It is therefore crucial to rewrite this part, supported by examples.
Answer: We thank the reviewer for making us aware of this important point. We modified the text accordingly. Please see:
- Lines 143-152: “Under physiological conditions, binding of tau to microtubules is a highly dynamic process that is dependent on several factors, including tau isoforms, mutations, post-translational modifications, but also the method used to determine the interaction between tau and microtubules [23]. Concerning the influence of post-translational modifications, this binding ability is most prominently regulated by tau’s phosphorylation state [35,48]. Kinase-mediated phosphorylation of tau detaches the protein from microtubules and subsequently causes their depolymerization, whereas phosphatases de-phosphorylate tau and retain the binding ability to microtubules [49]. Especially the phosphorylation of the KXGS motifs within the microtubule-binding domain has been shown to strongly reduce the binding ability of tau to microtubules [36,50].”
- Lines 183-211: “Hyperphosphorylation of tau is most likely to result from an imbalance in the activities of specific tau kinases and phosphatases, causing an increased rate of tau phosphorylation and/ or decreased rate of de-phosphorylation [66]. Consequently, tau hyperphosphorylation reduces its binding affinity to microtubules, thereby induces a loss of tau’s normal microtubule-stabilizing function [67,68], and thus causes microtubule depolymerization [49]. Specifically, the phosphorylation of KXGS motifs within the microtubule-binding domain (in particular S262) and S214 within the flanking region of tau have been described to strongly decrease the affinity of tau for microtubules [69,70]. Furthermore, in vitro studies demonstrated that phosphorylation of T231 within the flanking region also contributes to the reduced binding of tau to microtubules [71]. The detachment of tau from microtubules subsequently leads to an abnormal increase of free unbound tau in the cytosol [72]. This higher cytosolic concentration may render tau substantially more likely to undergo misfolding. Thereafter, as an early pathological event, non-fibrillar tau deposits, referred to as pre-tangles, are formed. Following steps comprise conformational changes leading to the generation of PHFs. This transition from pre-tangles to PHFs includes the formation of characteristic β-sheet like structures [11,73]. Precisely, the hexapeptide motifs PHF6 and PHF6* located in the second and third microtubule-binding repeats exhibit a high β-sheet propensity, and are supposed to promote abnormal tau aggregation in vitro and in cell and animal models [22,23,26]. Finally, PHFs further self-assemble to form more organized aggregates, and eventually develop insoluble NFTs inside neurons. The following sequestration of NFTs together with compromised cytoskeleton dynamics impairs normal axonal transport, and hence contributes to synaptic dysfunction and neurodegeneration [74,75]. In addition, alterations of tau itself, such as mutations in the MAPT gene, can also contribute to tau aggregation. For instance, in the tau mutations P301L, P301S and ΔK280 that are found in frontotemporal dementia with parkinsonism-17 (FTDP-17) the hexapeptide motif PHF6* is present. As a result of this enhanced β-sheet propensity, tau with these mutations tends to have a decreased affinity for microtubules and an increased ability to assemble into filaments, thus promoting tau aggregation [22,30,76]. To point out, even though phosphorylation of S262 and S214 strongly prevents the attachment of tau to microtubules, phosphorylation of these sites tends to inhibit PHF formation [21].].”
5/ - p4 lines 168-182: the authors here describe the involvement of tau hyperphosphorylation in tauopathies. However, with nearly 80 phosphorylabl sites, it is important to show and explain which ones are extremely important for the "loss of normal microtubule-stabilizing fraction" (l172), and in particular which sites are important for the generation of PHFs / NFTs.
Answer: We agree. Additional information were added. Please see the answer to the previous question and the text lines 183-211.
6/ - p5-6 lines 210-264: descriptions of the functioning of the mitochondria and its role as a factory to produce energy are obviously necessary for the novice, and the authors have carried out a very advanced editorial work. Perhaps a little too precisely, in particular in the passage devoted to the mechanisms of production of oxygenated species in which we do not see very well where the authors are coming from. We have to wait for the last sentence (lines 262-264) to understand that this whole description is important in neurodegenerative diseases (by the way, really all? Probably not, and only concerns those for which a dysfunction of the mitochondria is proven!) . Consequently, the reader remains a little unsatisfied since the authors do not describe the physiopathological studies illustrating their remarks. It would therefore be desirable to rewrite this part and rely on examples of studies.
Answer: We agree with the referee. The purpose of this section (3. Mitochondria) was to introduce the different aspects of mitochondrial function that are subsequently discussed in the section 4 (Mitochondria: Target of tau), in order to give key information for novices in the field. We precise this point in lines 260-262:
“In the following paragraphs, we will introduce key aspects of mitochondrial physiology that are relevant to understand the deleterious impact of abnormal tau on this paramount organelle (described in sections 4 and 5).”
Details about ROS generation by mitochondria were removed from the text. The role of mitochondrial dysfunction in aging and Alzheimer’s disease was the focus of other reviews from our group and illustrated with concrete examples (see Grimm A. et al, Biogerontol 2016, and Grimm A., et al, J Neurochem 2017). Therefore, this point was not extensively described in the present manuscript, which mainly aims to focus on the impact of tau on mitochondria. Nevertheless, we shortly discuss the role of mitochondria-derived ROS on Aβ generation and tau phosphorylation. Please see:
- Lines 309-315: “Namely, mitochondria-derived oxidative stress was proposed to be a causative factor for Aβ and tau pathology [1,98,100]. Indeed, Aβ load was increased in cells and mice that produced more ROS due to a mitochondrial Complex I inhibition/ deficiency [101]. Similarly, increased levels of tau and tau phosphorylation (at S396, S404, T205, T231) were observed in mice lacking the detoxifying enzyme superoxide dismutase 2 (SOD2) [102]. This suggests that mitochondrial dysfunction, more precisely mitochondria-derived ROS, might be involved in the pathogenesis of tauopathies.”
7/ - p6-9 lines 265-358: the description of the dynamics of mitochondria approached both at the protein and physiological level is extremely interesting and I think complete. However, the authors should consider this description in neurophysiopathological context: what are the consequences of ROS on the formation of PHF / NFT for example? Once again, this is not illustrated enough of a concrete example (e.g. studies carried out on cell lines, in vivo, patient cohort ...) and it is only towards the end of this part that we learns all the interest of the description in neurodegenerative diseases.
Answer: We thank the referee for this comment. Please see our answer to point 6 and lines 309-315.
8/ - p6 line 360-374: what is the interest of this section in the tau-mitochondria subject discussed here?
Answer: Please see our answer to point 10.
9/ - p12-13: Table 1 presenting the points implicating the tau protein in the dysfunction of the mitochondria is very useful. The authors also describe some of the most important points. Suddenly, this table could perhaps be moved before the authors' text, indicating the important points that will be taken up by the authors.
Answer: We agree with the referee. The table was moved before the text.
10/ - p13 paragraph “5. New insight ...”: I finally understand why the authors wrote above a part devoted to the ER-mitochondria coupling. In view of the short length of the previous paragraph (p9 lines 359-374), the authors could move it here as an introduction to this part 5.
Answer: We agree with this suggestion and we moved the introduction to this part accordingly. Please see lines 635-649.
11/ - p15: where is the comment for the figure 3 in the manuscript? In addition, this schema which tries to summarize the different targets of tau in the 5 processes involving the mitochondria (transport, dynamics, etc.) is quite difficult to understand: the authors should specify point by point whether tau has a promoting or disrupting role (maybe with another symbol for tau in the picture?) on the illustrated regulation (directly in the schema, and not only in the legend of figure). Also, it is difficult to understand from this rather complex schema the relationship between tau protein, the dysfunction of mitochondria and the appearance of the neurodegenerative disease, what about the effect of the hyperphosphorylation of tau, of possible mutations, of the ratio of expression between the 3R and 4R isoforms, etc ... in the disease? It is indeed difficult to synthesize all the points raised in the review by such a schema.
Answer: We thank the reviewer for this constructive comment. A key was added on Figure 3 to distinguish promoting and disrupting roles of tau. We agree that it is difficult to summarize in one schema the complex relationship between abnormal tau and mitochondria. That is why we specify in the figure legend that “the effects illustrated here may be different according to tau models (phospho-tau versus truncated tau)”, line 574-575. Also in the text, we precise that not all but “Important mitochondrial impairments observed in the presence of abnormal tau are summarized in Table 1 and Figure 3.” (lines 418-419). Finally, in order to link tau-induced mitochondrial dysfunction and neurodegeneration, a fourth figure was added. Please see Figure 4 and corresponding caption, lines 710-714.
Reviewer 4 Report
The review provides a clear overview on tau function in physiological and pathophysiological conditions. It is very well written and the structure of the review is logical and easy to follow.
Points to improve/add:
- Targeting tau in AD is present in many clinical trials and this needs to be added
- How the potential drugs (in clinical trials) targeting tau affect mitochondria?
- A description of the role of tau in the immune cells is missing
- More work on human iPSC and organoids could be included
Minor points:
- the exclusive maternal lineage of mitochondria should be revisited
Author Response
Targeting tau in AD is present in many clinical trials and this needs to be added
How the potential drugs (in clinical trials) targeting tau affect mitochondria?
Answer: We agree with the referee. A statement about current therapeutic strategies against tau pathology, as well as the effects of drugs undergoing clinical trials on mitochondria were added. Please see lines 229-248:
“Current therapeutic strategies targeting tau consist in anti-aggregation agents (regulation of tau phosphorylation, inhibition of tau aggregation), tau passive immunotherapy, tau therapeutic vaccines, targeting of tau gene expression (antisense therapies) and therapeutic reduction of tau (reviewed in [83]). Strikingly, the effects of tau-targeting drugs on mitochondrial function remain under-investigated. Conversely, evidence showed that improving mitochondrial quality control via activation of mitophagy (removal of damaged mitochondria, see Section 3.2.) decreases tau pathology in different experimental models (reviewed in [84]). Namely, nicotinamide riboside supplementation (activator of mitophagy) was shown to decrease abnormal tau phosphorylation, neuroinflammation and cognitive impairments in AD transgenic animals [85,86]. Of note, mitophagy itself seems to play also a role in the regulation of the inflammatory response [87]. Further studies need to be performed to unravel the role of mitophagy in the reduction of tau pathology via modulation of neuroinflammation.
To our knowledge, only one of the tau-targeting agents undergoing clinical trials was shown to have also an effect on mitochondria [88]. Indeed, methylene blue (MB), also known as methylthionine chloride (MTC), is already an approved drug against malaria, and acts as a direct inhibitor of tau protein aggregation [83]. MTC was shown to counteract oxidative stress-induced mitochondrial damage, and to inhibit the monoamine oxidase A that is a source of reactive oxygen species (ROS)[88]. Despite the improvements observed in AD-related symptoms during phase II clinical trials, too many undesirable side effects were reported (e.g. dizziness, diarrhoea, painful urination) for the drug to be used (Clinical Trial Identifier, NCT00515333 and NCT00684944) [83].”
A description of the role of tau in the immune cells is missing
Answer: We thank the reviewer for this suggestion. More information were added. Please see lines 223-228: “Of note, increasing evidence has been linking tau pathology and neuroinflammation. Indeed, abnormal tau was associated with reactive microglia, as well as increased levels of pro-inflammatory cytokines (e.g. interleukin-1β) and complement proteins (reviewed in [81]). The chronic neuroinflammation may lead to synapsic loss and cognitive decline. Interestingly, microglial cells themselves seem to be involved in the spreading of tau pathology ([82]).
More work on human iPSC and organoids could be included
Answer: We thank the referee for this suggestion. As the aim of the review was to focus on the impact of tau on mitochondria, we realized that only few studies using IPSC/organoids were published about this specific topic (in “pure” tau models without amyloid component). Therefore, two new references were added and discussed. Please see Table 1 and lines 458-464:
“Impairments in mitochondrial axonal transport were also evident in induced pluripotent stem cells (IPSCs) derived from frontotemporal dementia (FTD) patients bearing the R406W tau mutation [124]. In these IPSCs induced into cerebral organoids, axonal mitochondria were less stationary and moved more in the retrograde direction, resulting in less mitochondria into the axon when compared to control cells. In line, anterograde axonal transport of mitochondria was significantly reduced in IPSC-derived neurons bearing the N279K and P301L tau mutation compared to controls [123].]”.
Minor points:
the exclusive maternal lineage of mitochondria should be revisited
Answer: We thank the referee for this comment. The following statement was added. Please see lines 251-253:
“Unlike other organelles, they are maternally inherited (although biparental inheritance of mitochondrial DNA was recently reported but still under debate [89,90]) and compartmentalized
Round 2
Reviewer 2 Report
The authors have shown a lot of efforts to improve the manuscript and this should be well appreciated. I found the authors have addressed all my comments carefully and in detail by adding more materials in the text and correcting figures. As a result, I now recommend the current form can be accepted for publication.